# Polarization Direction of Arrival Estimation Using Dual Algorithms Based on Time-Frequency Cross Terms

Shuai Shao , Aijun Liu , Xiuhong Wang * and Changjun Yu

School of Information Science and Engineering, Harbin Institute of Technology at Weihai, Weihai 264209, China; hit_shaoshuai@163.com (S.S.); liuaijun@hit.edu.cn (A.L.); yuchangjun@hit.edu.cn (C.Y.)
* Correspondence: xiuhongwang@hit.edu.cn

**Abstract:** In radar array signal processing, weak nonstationary polarization signal direction of arrival (DOA) estimation is a challenging and significant issue when both strong and weak nonstationary signals coexist. Time-frequency (T-F) analysis is an effective method to deal with nonstationary signals. In the last decade, spatial time-frequency distributions (STFDs) have been proposed for multiple dual-polarization antenna arrays and efficaciously used for nonstationary signal DOA estimation. In this article, we introduced a novelty means for estimating weak nonstationary polarization signal DOA by utilizing spatial polarimetric time-frequency distributions (SPTFDs) of cross terms when there are strong nonstationary polarization interference signals and additive Gaussian white noise. The cross terms' SPTFDs are considered via a replaceability matrix for data covariance in multiple signal classification (MUSIC) and the estimation of signal parameters using the rotational invariance technique (ESPRIT). Combining the STFDs of cross terms with polarization information about weak nonstationary signals improves signal DOA estimation accuracy. The combined MUSIC and ESPRIT are used in the algorithm to further ensure the success probability and accuracy of DOA estimation. Through simulation analyses, the proposed algorithm is more suitable for the required application scenarios than other algorithms and is superior to other algorithms.

**Keywords:** time-frequency analysis; polarization direction of arrival estimation; cross terms; spatial polarimetric time-frequency distribution

## 1. Introduction

Time-frequency analysis has been introduced for nonstationary signal analysis in many fields, for instance, machine monitoring, speech, the automotive industry and biomedicine [1–3]. In the last many years, the spatial dimension has been combined with the time and frequency dimensions, including quadratic and higher-order time-frequency distributions (TFDs), which resulted in spatial time-frequency distributions (STFDs) development for nonstationary array signal processing [4–7]. The relation between the sensor data TFDs and the source TFDs is confirmed by the steering mixing matrix. The sensor data TFDs are similar to the traditional data covariance matrix in array signal processing [8,9]. This similarity has a guaranteed subspace algorithm to use the instantaneous frequency of the source for signal DOA estimation. The MUSIC [10] and ESPRIT [11] algorithms based on STFDs are superior to their counterparts based on the data covariance matrix when utilized for the DOA estimation of nonstationary signals [4,12].

Polarization is generally not only used in satellite communications and wireless but also in various radar array signal processing methods [13–16]. Antenna and target polarization characteristics are generally applied in synthetic aperture radar (SAR) systems and remote sensing [17,18]. Polarization information is included in airborne and spaceborne platforms and meteorological radars [19]. Moreover, for target identification in the clutter, polarization plays a significant role [20]. Polarization information has also been applied for polarization sensitivity antenna arrays to promote signal parameter estimation performance, which primarily includes time of arrival and DOA [21–23].

The two significant fields of time-frequency analysis and polarization signal processing have rarely cooperated in the same platform, despite the abundant research executed in each field separately [24,25]. In this study, we propose SPTFDs based on cross terms for dual-polarization arrays, where the signal T-F and polarized characteristics are simultaneously sufficiently applied. The virtues of the SPTFDs based on cross terms are confirmed by far-field narrow-band point signals incoming to the array. The polarized information increases an additional degree of freedom of the STFDs, resulting in ameliorative signal spatial resolution and DOA estimation accuracy.

When strong and weak nonstationary signals simultaneously arise, the weak target signal auto terms are submerged into the background noise in the time-frequency analysis. They are difficult to extract and use directly. The cross terms also contain weak target signal information [26,27]. Unlike the suppression of cross terms [28,29], the SPTFD matrix is constructed by the T-F points of cross terms to estimate the weak target signal DOA in this algorithm [30,31].

SPTFDs are utilized for defining the polarimetric time-frequency MUSIC (PTF-MUSIC) algorithm, combining the source T-F and polarized characteristics for nonstationary signal polarization DOA estimation. The PTF-MUSIC algorithm is superior to the MUSIC algorithms that merely consider polarization or T-F peculiarities. In addition, an ESPRIT-like algorithm is proposed in [32]. The algorithm in this paper combines the characteristics of the MUSIC and the ESPRIT algorithms to ensure success rate and accuracy. Rough estimation and fine estimation are combined to ensure that the algorithm has a high success rate and relatively accurate target direction estimation when strong and weak nonstationary signals simultaneously arise.

This paper mainly solves the problem of the poor DOA estimation performance of weak nonstationary polarized signals in the presence of strong nonstationary polarized interference signals and additive white Gaussian noise. Polarization direction of arrival estimation is proposed using dual algorithms based on time-frequency cross terms. Instead of suppressing cross terms, the method uses cross terms. From a novel perspective, the cross term contains weak target signal information and has an obvious antidiagonal matrix structure which is easy to extract.

This paper includes the following sections. Section 2 describes polarization modeling and discusses SPTFDs. Section 3 studies time-frequency point selection. Polarization DOA estimation using dual algorithms based on T-F cross terms is introduced in Section 4. Section 5 discusses the issues of spatial polarimetric correlations. The simulation analyses are offered in Section 6, which demonstrates the proposed algorithm's effectiveness.

## 2. Spatial Polarimetric Time-Frequency Distribution

### 2.1. Polarization Modeling

In Figure 1, the electric field of transverse electromagnetic (TEM) waves incoming to the array can be expressed as

$$
\begin{aligned}
\mathbf{E}(t) = E_\theta(t)\boldsymbol{\theta} + E_\phi(t)\boldsymbol{\phi} = &\left[E_\theta(t)\cos(\theta)\cos(\phi) - E_\phi(t)\sin(\phi)\right]\mathbf{x} \\
&+ \left[E_\theta(t)\cos(\theta)\sin(\phi) + E_\phi(t)\cos(\phi)\right]\mathbf{y} + E_\theta(t)\sin(\theta)\mathbf{z},
\end{aligned}
\tag{1}
$$

where $\boldsymbol{\phi}$ and $\boldsymbol{\theta}$ are the azimuth and elevation spherical unit vectors of the signals. The $\mathbf{x}$, $\mathbf{y}$ and $\mathbf{z}$ are, respectively, rectangular coordinate unit vectors along the x, y and z directions. For generality and succinctness, the signal and arrays are assumed in the x-y plane and the y-z plane, respectively. Then, $\theta = 90°(\boldsymbol{\theta} = -\mathbf{z})$, and

$$
\mathbf{E}(t) = -E_\phi(t)\sin(\phi)\mathbf{x} + E_\phi(t)\cos(\phi)\mathbf{y} + E_\theta(t)\mathbf{z}.
\tag{2}
$$

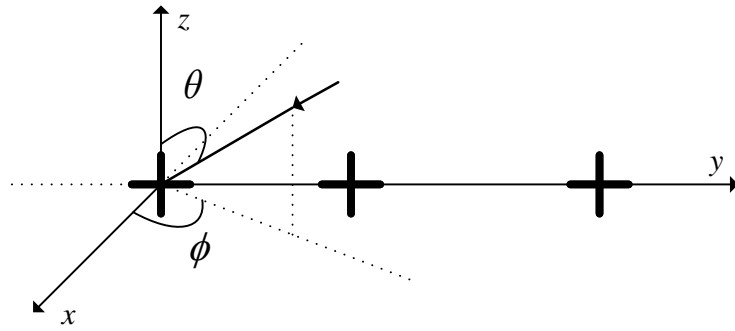

**Figure 1.** Dual-polarized antenna array.

$s(t)$ is expressed as the source amplitude measured by the receiving reference antenna sensor. The polarization auxiliary angle is $\gamma \in [0, \pi/2]$ and the polarization phase difference is $\eta \in (-\pi, \pi]$. $s^{[v]}(t)$ and $s^{[h]}(t)$ are the vertical and horizontal polarization components of the signal source. $E_\theta(t)$ and $E_\phi(t)$ can be expressed by spherical fields as

$$E_\theta(t) = s^{[v]}(t) = s(t)\cos(\gamma), E_\phi(t) = s^{[h]}(t) = s(t)\sin(\gamma)e^{j\eta}. \tag{3}$$

A signal is linearly polarized if $\eta = 0°$ or $\eta = 180°$. Bringing (3) into (2) yields

$$\mathbf{E}(t) = s(t)[-\cos(\gamma)\sin(\phi)\mathbf{x} + \cos(\phi)\sin(\gamma)e^{j\eta}\mathbf{y} + \cos(\gamma)\mathbf{z}]. \tag{4}$$

The $N$ signals are assumed entering on the $M$ dual-polarization antenna array. The $n$th source's dual-polarized components are

$$s_n^{[v]}(t) = s_n(t)\cos(\gamma_n) = c_{nv}s_n(t), s_n^{[h]}(t) = s_n(t)\sin(\gamma_n)e^{j\eta_n} = c_{nh}s_n(t), \tag{5}$$

where the parameters $c_{nv} = \cos(\gamma_n)$ and $c_{nh} = \sin(\gamma_n)e^{j\eta_n}$ describe the vertical and horizontal polarization coefficients. The signal source incident on the $m$th dual-polarization antenna sensor expresses

$$\begin{aligned}
\underline{y}_m(t) &= \left[y_m^{[v]}(t), y_m^{[h]}(t)\right]^{\mathrm{T}} = \sum_{n=1}^{N}\left[a_{nm}^{[v]}\mathbf{E}_n \cdot \mathbf{z}, a_{nm}^{[h]}\mathbf{E}_n \cdot \mathbf{y}\right]^{\mathrm{T}} \\
&= \sum_{n=1}^{N}\left[a_{nm}^{[v]}s_n^{[v]}(t), a_{nm}^{[h]}s_n^{[h]}(t)\cos(\phi_n)\right]^{\mathrm{T}},
\end{aligned} \tag{6}$$

where "$\cdot$" is the dot product and $\mathbf{E}_n$ is the $n$th signal source electric field vector. $a_{nm}^{[v]}$ and $a_{nm}^{[h]}$ are the $m$th elements of the vertical and horizontal polarization steering vectors, $\mathbf{a}^{[v]}(\phi_n)$ and $\mathbf{a}^{[h]}(\phi_n)$, respectively. It is assumed that $\mathbf{a}^{[v]}(\phi)$ and $\mathbf{a}^{[h]}(\phi)$ are normalized ($\left\|\mathbf{a}^{[v]}(\phi)\right\|^2 = \left\|\mathbf{a}^{[h]}(\phi)\right\|^2 = M$) and the array has been calibrated. In this paper, the $\cos(\phi_n)$ terms of the horizontal polarization array manifold can be introduced in the array calibration. Therefore, they will not be considered. Equation (6) is simplified as

$$\begin{aligned}
\underline{y}_m(t) &= \left[a_{nm}^{[v]}s_n^{[v]}(t), a_{nm}^{[h]}s_n^{[h]}(t)\right]^{\mathrm{T}} = s_n(t)\left(\begin{bmatrix} a_{nm}^{[v]} & a_{nm}^{[h]} \end{bmatrix}^{\mathrm{T}} \odot \begin{bmatrix} c_{nv} & c_{nh} \end{bmatrix}^{\mathrm{T}}\right) \\
&= s_n(t)\mathbf{a}_{nm} \odot \mathbf{c}_n,
\end{aligned} \tag{7}$$

where $\mathbf{c}_n = [c_{nv}, c_{nh}]^{\mathrm{T}} = \left[\cos(\gamma_n), \sin(\gamma_n)e^{j\eta_n}\right]^{\mathrm{T}}$ expresses the $n$th signal polarization characteristic parameter.

### 2.2. Spatial Polarimetric Time-Frequency Distributions

The fundaments of STFDs are discussed. The Cohen classes, especially the Wigner–Ville distribution (WVD) and its peculiarities, are examined. The WVD formulation of a signal $x(t)$ is given by

$$D_{xx}(t,f) = \iint \varphi(t - u, \tau) x\left(u + \frac{\tau}{2}\right) \times x^*\left(u - \frac{\tau}{2}\right) e^{-j2\pi f\tau} du d\tau, \tag{8}$$

where $t$ and $f$ are, respectively, the time and frequency indexes, j is the imaginary unit and * implies complex conjugate. The kernel $\varphi(t, \tau)$ is a function of the time and lag variables and uniquely defines the TFDs. In this paper, all the integrals are from $-\infty$ to $\infty$.

With the $M$ dual-polarized antennas array, the data vector of each polarization is expressed as

$$\mathbf{x}^{[i]}(t) = \left[x_1^{[i]}(t), x_2^{[i]}(t), \ldots, x_M^{[i]}(t)\right]^{\mathrm{T}} = \mathbf{y}^{[i]}(t) + \mathbf{n}^{[i]}(t)$$
$$= \mathbf{A}^{[i]}(\mathbf{\Phi})\mathbf{s}^{[i]}(t) + \mathbf{n}^{[i]}(t)(i = v \text{ or } h), \tag{9}$$

where $\mathbf{A}^{[i]}(\mathbf{\Phi}) = [\mathbf{a}_1, \mathbf{a}_2, \ldots, \mathbf{a}_M] = [\mathbf{a}(\phi_1), \mathbf{a}(\phi_2), \ldots, (\phi_M)]$ and $\mathbf{a}(\phi_n) = -2\pi(n-1)d\sin\phi/\lambda$. The array is a uniform linear array, and the array manifold vector is $\mathbf{A}$, where each element represents the wavepath difference reached by the electromagnetic wave. Therefore, the DOA of the signal can be derived from the array manifold. The autopolarization STFDs matrix of the data vector is defined as

$$\mathbf{D}_{\mathbf{x}^{[i]}\mathbf{x}^{[i]}}(t,f) = \iint \varphi(t-u,\tau)\mathbf{x}^{[i]}\left(u + \frac{\tau}{2}\right) \times \left(\mathbf{x}^{[i]}\left(u - \frac{\tau}{2}\right)\right)^{\mathrm{H}} e^{-j2\pi f\tau} du d\tau, \tag{10}$$

where H is conjugate transpose. In the noise-free circumstance, (10) can be described as

$$\mathbf{D}_{\mathbf{x}^{[i]}\mathbf{x}^{[i]}}(t,f) = \mathbf{A}^{[i]}(\mathbf{\Phi})\mathbf{D}_{\mathbf{s}^{[i]}\mathbf{s}^{[i]}}(t,f)\left(\mathbf{A}^{[i]}(\mathbf{\Phi})\right)^{\mathrm{H}}. \tag{11}$$

Similarly, the cross-polarization STFDs matrix between two different polarization data vectors can be described as

$$\mathbf{D}_{\mathbf{x}^{[i]}\mathbf{x}^{[j]}}(t,f) = \iint \varphi(t-u,\tau)\mathbf{x}^{[i]}\left(u + \frac{\tau}{2}\right) \times \left(\mathbf{x}^{[j]}\left(u - \frac{\tau}{2}\right)\right)^{\mathrm{H}} e^{-j2\pi f\tau} du d\tau, \tag{12}$$

which becomes

$$\mathbf{D}_{\mathbf{x}^{[i]}\mathbf{x}^{[j]}}(t,f) = \mathbf{A}^{[i]}(\mathbf{\Phi})\mathbf{D}_{\mathbf{s}^{[i]}\mathbf{s}^{[j]}}(t,f)\left(\mathbf{A}^{[j]}(\mathbf{\Phi})\right)^{\mathrm{H}}. \tag{13}$$

On the basis of (9), the extended data vector of both polarizations can be established thus:

$$\mathbf{x}(t) = \begin{bmatrix} \mathbf{x}^{[v]}(t) \\ \mathbf{x}^{[h]}(t) \end{bmatrix} = \begin{bmatrix} \mathbf{A}^{[v]}(\mathbf{\Phi}) & \mathbf{0} \\ \mathbf{0} & \mathbf{A}^{[h]}(\mathbf{\Phi}) \end{bmatrix} \begin{bmatrix} \mathbf{s}^{[v]}(t) \\ \mathbf{s}^{[h]}(t) \end{bmatrix} + \begin{bmatrix} \mathbf{n}^{[v]}(t) \\ \mathbf{n}^{[h]}(t) \end{bmatrix}$$
$$= \begin{bmatrix} \mathbf{A}^{[v]}(\mathbf{\Phi}) & \mathbf{0} \\ \mathbf{0} & \mathbf{A}^{[h]}(\mathbf{\Phi}) \end{bmatrix} \begin{bmatrix} \mathbf{Q}^{[v]} \\ \mathbf{Q}^{[h]} \end{bmatrix} \mathbf{s}(t) + \begin{bmatrix} \mathbf{n}^{[v]}(t) \\ \mathbf{n}^{[h]}(t) \end{bmatrix} = \mathbf{B}(\mathbf{\Phi})\mathbf{Q}\mathbf{s}(t) + \mathbf{n}(t), \tag{14}$$

where

$$\mathbf{B}(\mathbf{\Phi}) = \begin{bmatrix} \mathbf{A}^{[v]}(\mathbf{\Phi}) & \mathbf{0} \\ \mathbf{0} & \mathbf{A}^{[h]}(\mathbf{\Phi}) \end{bmatrix}, \tag{15}$$

is a block diagonal matrix and

$$\mathbf{Q} = \begin{bmatrix} \mathbf{Q}^{[v]} \\ \mathbf{Q}^{[h]} \end{bmatrix}, \tag{16}$$

is the signal polarization characteristic vector, where

$$\mathbf{q}^{[v]} = [\cos(\gamma_1), \ldots, \cos(\gamma_N)]^{\mathrm{T}}, \mathbf{Q}^{[v]} = \mathrm{diag}\left(\mathbf{q}^{[v]}\right), \tag{17}$$

$$\mathbf{q}^{[h]} = \left[\sin(\gamma_1)e^{j\eta_1}, \ldots, \sin(\gamma_N)e^{j\eta_N}\right]^{\mathrm{T}}, \mathbf{Q}^{[h]} = \mathrm{diag}\left(\mathbf{q}^{[h]}\right). \tag{18}$$

Then,

$$\mathbf{B}(\boldsymbol{\Phi})\mathbf{Q} = \begin{bmatrix} \mathbf{a}^{[v]}(\phi_1)c_{1v} & \cdots & \mathbf{a}^{[v]}(\phi_n)c_{Nv} \\ \mathbf{a}^{[h]}(\phi_1)c_{1h} & \cdots & \mathbf{a}^{[h]}(\phi_n)c_{Nh} \end{bmatrix} = \begin{bmatrix} \widetilde{\mathbf{a}}(\phi_1) & \cdots & \widetilde{\mathbf{a}}(\phi_N) \end{bmatrix}. \tag{19}$$

The matrix (19) can be considered as the extended mixing steering matrix, with $\widetilde{\mathbf{a}}(\phi_n)$ representing the spatial polarimetric extended characteristic vector. The spatial polarimetric characteristic extended vector of the $n$th signal is

$$\widetilde{\mathbf{a}}(\phi_n) = \begin{bmatrix} \mathbf{a}^{[v]}(\phi_n)\cos(\gamma_n) \\ \mathbf{a}^{[h]}(\phi_n)\sin(\gamma_n)e^{j\eta_n} \end{bmatrix}. \tag{20}$$

Compared to a single-polarization array, the dual-polarization array doubles the vectors' spatial dimensionality. It is now possible to combine the spatial, polarized and T-F information of the signals incoming to the array. For the dual-polarized data vector, the SPTFDs matrix is described as

$$\mathbf{D}_{\mathbf{xx}}(t, f) = \iint \varphi(t - u, \tau)\mathbf{x}\left(u + \frac{\tau}{2}\right) \times \mathbf{x}^{\mathrm{H}}\left(u - \frac{\tau}{2}\right)e^{-j2\pi f\tau}\mathrm{d}u\mathrm{d}\tau. \tag{21}$$

This matrix serves as a general framework where classical issues in radar array processing can be resolved, for instance, DOA estimation. When the noise effect is neglectful, the SPTFDs matrix is in connection with the signal TFDs matrix via

$$\mathbf{D}_{\mathbf{xx}}(t, f) = \mathbf{B}(\boldsymbol{\Phi})\mathbf{Q}\mathbf{D}_{\mathbf{ss}}(t, f)\mathbf{Q}^{\mathrm{H}}\mathbf{B}^{\mathrm{H}}(\boldsymbol{\Phi}). \tag{22}$$

## 3. Time-Frequency Point Selection

The advantages of polarization DOA estimation based on T-F cross terms can only be reflected if reasonable T-F points are selected for SPTFDs matrix establishment.

### 3.1. SPTFDs Properties

Consider an $N \times M$ whitening matrix $\mathbf{W}$ with unitary mixing matrix $\mathbf{U} = \mathbf{WA}$ and

$$(\mathbf{WA})(\mathbf{WA})^{\mathrm{H}} = \mathbf{U}\mathbf{U}^{\mathrm{H}} = \mathbf{I}, \tag{23}$$

where $\mathbf{I}$ denotes the identity matrix. Pre- and postmultiplying $\mathbf{D}_{\mathbf{xx}}(t, f)$ by $\mathbf{W}$ results in the whitened matrix, defined as

$$\underline{\mathbf{D}}_{\mathbf{xx}}(t, f) = \mathbf{W}\mathbf{D}_{\mathbf{xx}}(t, f)\mathbf{W}^{\mathrm{H}} = \mathbf{U}\mathbf{D}_{\mathbf{ss}}(t, f)\mathbf{U}^{\mathrm{H}}, \tag{24}$$

where the middle equation is from (22) and (23). The whitening step results in a linear model. The whitening matrix is calculated as an inverse square root from the data covariance matrix [33] or the STFDs matrix [4]. The whitening matrix is based on independent signals. However, the SPTFDs matrix needs not the above condition. The SPTFDs matrix in (22) and (24) guarantees the effective subspace algorithms to conduct a class of issues, for instance, blind sources separation and high-resolution DOA estimation [34].

### 3.2. Time-Frequency Point Properties and Categorization

The SPTFDs framework requires the signal T-F characteristics to satisfy one or two conditions with adequate difference:

(1) The T-F points correspond to auto terms. That is, if $D_{s_i s_j}(t,f) = (\mathbf{D_{ss}}(t,f))_{ij}$, then

$$D_{s_i s_j}(t_n, f_n) = \delta_{i,j} D_{i,j,n}. \tag{25}$$

There is at least an $n$th T-F point for each $N$ signal, such that $D_{i,j,n} \neq 0$. $\delta_{i,j}$ represents the Kronecker delta. $D_{i,j,n}$ is the value of the SPTFDs between the signals $s_i$ and $s_j$ (or $s_i$) at the T-F point.

(2) The T-F points correspond to cross terms. That is,

$$D_{s_i s_j}(t_n, f_n) = (1 - \delta_{i,j}) D_{i,j,n}. \tag{26}$$

In the SPTFDs framework, the two assumptions imply that the signal T-F signatures could not overlap strongly. For DOA estimation algorithms, the adequately different signatures represent the signal's known discriminating property.

The virtues of the T-F-based polarization DOA estimation algorithm may be merely reflected when reasonable T-F points are selected for SPTFDs matrix establishment. The algorithm's key point is how to select reasonable T-F points. This paper's objective is the polarization DOA estimation of the weak nonstationary signal from strong nonstationary signal interference. The signal source SPTFDs matrix is

$$\mathbf{D_{ss}}(t,f) = \begin{pmatrix} \mathbf{D_{s_1 s_1}}(t,f) & \mathbf{D_{s_1 s_2}}(t,f) \\ \mathbf{D_{s_2 s_1}}(t,f) & \mathbf{D_{s_2 s_2}}(t,f) \end{pmatrix}. \tag{27}$$

The $\mathbf{s}_1(t)$ and $\mathbf{s}_2(t)$ are two polarization linear frequency modulation signals. The four types of T-F points are described. The first type of T-F point is merely in connection with auto terms. The TFDs matrix possesses a rank-one diagonal structure for those points. The second type of T-F point is merely in connection with cross terms. The TFDs matrix possesses an off-diagonal (a matrix is said to be off-diagonal if its diagonal entries are zeros) structure for those points. Because their diagonal elements are zeros, the matrix is considered to be off-diagonal. The third type of T-F point is in connection with both auto terms and cross terms, however. The auto and cross terms do not exist in the fourth type of T-F point. The TFDs matrices of the third and fourth types of T-F points have no clear algebraic structure for those points. Therefore, they can not be straightforwardly utilized.

The diagonal and off-diagonal structures are frequently destroyed when the signals are mixed. For DOA estimation, the first and second types of T-F points are effective and important. Because there is effect in this situation, the third and fourth T-F points should be abnegated. When there are both strong and weak nonstationary signals, in this paper, the cross terms of SPTFDs are exploited to estimate weak nonstationary signal DOA. Because of the high outstanding off-diagonal algebraic structure of the second type of T-F points, a novelty polarization DOA estimation algorithm will exploit the SPTFDs matrix decomposition technology of T-F cross terms. The relationship between cross terms' SPTFDs and weak signal DOA is derived as follows. After cross terms extraction, the source time-frequency distribution matrix is

$$\mathbf{D_{ssc}}(t,f) = \begin{pmatrix} \mathbf{0} & \mathbf{D_{s_1 s_2}}(t,f) \\ \mathbf{D_{s_2 s_1}}(t,f) & \mathbf{0} \end{pmatrix}. \tag{28}$$

Further, the SPTFDs matrix of cross terms is obtained as follows.

$$\mathbf{D_{xxc}}(t,f) = \begin{bmatrix} \widetilde{\mathbf{a}}(\phi_1) & \widetilde{\mathbf{a}}(\phi_2) \end{bmatrix} \times \begin{pmatrix} \mathbf{0} & \mathbf{D_{s_1 s_2}}(t,f) \\ \mathbf{D_{s_2 s_1}}(t,f) & \mathbf{0} \end{pmatrix} \begin{bmatrix} \widetilde{\mathbf{a}}^{\mathrm{H}}(\phi_1) \\ \widetilde{\mathbf{a}}^{\mathrm{H}}(\phi_2) \end{bmatrix}$$
$$= \widetilde{\mathbf{a}}(\phi_2) \mathbf{D_{s_2 s_1}}(t,f) \widetilde{\mathbf{a}}^{\mathrm{H}}(\phi_1) + \widetilde{\mathbf{a}}(\phi_1) \mathbf{D_{s_1 s_2}}(t,f) \widetilde{\mathbf{a}}^{\mathrm{H}}(\phi_2). \tag{29}$$

It can be seen from Equation (29) that the SPTFD matrix of cross terms still contains the available DOA information of a weak target signal. The results provide theoretical support for applying the SPTFDs matrix of cross terms to DOA estimation.

### 3.3. Time-Frequency Points Selection Procedures

By selecting cross term T-F points, the data cross-source TFDs will have the following structure:

$$\underline{\mathbf{D}}_{\mathbf{xxc}}(t,f) = \mathbf{U}\mathbf{D}_{\mathbf{ssc}}(t,f)\mathbf{U}^{\mathrm{H}}, \tag{30}$$

where $\mathbf{D}_{\mathbf{ssc}}(t,f)$ is antidiagonal. The missing unitary matrix $\mathbf{U}$ is uniquely retrieved by the joint antidiagonalization (JAD) of a combined set $\{\underline{\mathbf{D}}_{\mathbf{xxc}}(t_i, f_i)|i = 1, \ldots, q\}$ of $q$ cross-source TFDs matrices. The JAD is explained by first noting that the problem of the antidiagonalization of a single $N \times N$ matrix $\mathbf{N}$ is equivalent (this is due to the fact that the Frobenius norm of a matrix is constant under unitary transform) to the maximization of the criterion

$$C(\mathbf{N}, \mathbf{V}) \stackrel{\text{def}}{=} -\sum_{i=1}^{N} \left|\mathbf{v}_i^H \mathbf{N}\mathbf{v}_i\right|^2, \tag{31}$$

over the set of unitary matrices $\mathbf{V} = [\mathbf{v}_1, \cdots, \mathbf{v}_n]$. Hence, the JAD of a set $\{\mathbf{N}_k \mid k = 1, \ldots, q\}$ of $q$ matrices is defined as the maximization of the JAD criterion,

$$C(\mathbf{V}) \stackrel{\text{def}}{=} \sum_{k=1}^{q} C(\mathbf{N}_k, \mathbf{V}) = -\sum_{k=1}^{q}\sum_{i=1}^{N} \left|\mathbf{v}_i^H \mathbf{N}_k \mathbf{v}_i\right|^2, \tag{32}$$

under the same unitary constraint. A Jacobi-like algorithm has been derived for the maximization of the JAD criterion (32). The specific process is described below. (1) Determine the whitening matrix $\hat{\mathbf{W}}$ from the eigendecomposition of an estimate of the data covariance matrix. (2) Compute the TFDs of the array output according to (22). (3) Select a set of T-F points (usually corresponding to the high amplitude points of the signal T-F transform), and then distinguish between auto term and cross term points using the above selection procedure. (4) Determine the unitary matrix $\hat{\mathbf{U}}$ by maximizing the JD/JAD criterion applied to the whitened TFD matrices computed at the selected T-F points. (5) Obtain an estimate of the mixture matrix $\hat{\mathbf{A}}$ as $\hat{\mathbf{A}} = \hat{\mathbf{W}}^{\#}\hat{\mathbf{U}}$, where the superscript # denotes the pseudoinverse, and an estimate of the source signals $\hat{\mathbf{s}}(t)$ as $\hat{\mathbf{s}}(t) = \hat{\mathbf{U}}^H \mathbf{W}\mathbf{x}(t)$.

The success of the JD or JAD of TFDs matrices in determining the unitary matrix $\mathbf{U}$ depends strongly on the correct selection of the auto and cross terms points. It is crucial to have a selection procedure that is able to distinguish between auto and cross terms points based only on the TFDs matrices of the observations. Here, we propose a selection approach that exploits the antidiagonal structure of the cross-source TFDs matrices. More precisely, we have

$$\text{Trace}(\underline{\mathbf{D}}_{\mathbf{xxc}}(t,f)) = \text{Trace}\left(\mathbf{U}\mathbf{D}_{\mathbf{ssc}}(t,f)\mathbf{U}^H\right) = \text{Trace}(\mathbf{D}_{\mathbf{ssc}}(t,f)) \approx 0. \tag{33}$$

The T-F points' automatic selection procedure is generally difficult, as described in the following. In a whitening case, a matrix's trace invariance for a unitary transform is utilized to judge the cross terms' existence. The matrices for the second type of T-F points require the following condition:

$$\frac{\text{trace}\left\{\underline{\mathbf{D}}_{\mathbf{xx}}\left(k_t, k_f\right)\right\}}{\left\|\underline{\mathbf{D}}_{\mathbf{xx}}\left(k_t, k_f\right)\right\|} < \varepsilon, \tag{34}$$

where trace{} is matrix trace, $\varepsilon$ is a positive small user-defined value and $\|\ \|$ is the Frobenius norm. The noise influence is further reduced by averaging the selected SPTFDs of cross terms.

Most procedures take advantage of a diagonal rank-one matrix for the auto terms. The emphasis of this paper is to make use of the off-diagonalization characteristic of the cross terms matrix. Firstly, cross terms not drowned by noise are extracted by the threshold value. Then, the diagonal element of the cross terms is close to 0 for extracting the cross terms further. The constructed SPTFDs are used for DOA estimation.

## 4. Polarimetric Time-Frequency DOA Estimation Using Dual Algorithms

The TF-MUSIC algorithm has recently been proposed for improving the spatial resolution of a good T-F peculiarity signal [35]. The algorithm in this paper provides the extension for applying TF-MUSIC based on cross terms to polarization arrays. Consider the spatial steering matrix

$$\mathbf{F}(\phi) = \frac{1}{\sqrt{M}} \begin{bmatrix} \mathbf{a}^{[v]}(\phi) & \mathbf{0} \\ \mathbf{0} & \mathbf{a}^{[h]}(\phi) \end{bmatrix}, \tag{35}$$

corresponding to the signal DOA. Because $\left\| \mathbf{a}^{[i]}(\phi) \right\|^2 = M$, $\mathbf{F}^{\mathrm{H}}(\phi)\mathbf{F}(\phi)$ represents the identity matrix. To search in the spatial and polarimetric joint domains, the spatial polarization vector is defined as:

$$\mathbf{f}(\phi, \mathbf{c}) = \frac{\mathbf{F}(\phi)\mathbf{c}}{\|\mathbf{F}(\phi)\mathbf{c}\|} = \mathbf{F}(\phi)\mathbf{c}, \tag{36}$$

where $\mathbf{c} = \begin{bmatrix} c_v & c_h \end{bmatrix}^{\mathrm{T}}$ represents a unit norm vector of the unknown polarization coefficient. In (36), $\|\mathbf{F}(\phi)\mathbf{c}\| = \left[ \mathbf{c}^H \mathbf{F}^H(\phi)\mathbf{F}(\phi)\mathbf{c} \right]^{1/2} = \left( \mathbf{c}^H \mathbf{c} \right)^{1/2} = 1$.

The PTF-MUSIC spectrum is provided via the following function:

$$P(\phi) = \left[ \min_{\mathbf{c}} \mathbf{f}^{\mathrm{H}}(\phi, \mathbf{c}) \mathbf{U}_n \mathbf{U}_n^{\mathrm{H}} \mathbf{f}(\phi, \mathbf{c}) \right]^{-1} = \left[ \min_{\mathbf{c}} \mathbf{c}^{\mathrm{H}} \mathbf{F}^{\mathrm{H}}(\phi) \mathbf{U}_n \mathbf{U}_n^{\mathrm{H}} \mathbf{F}(\phi) \mathbf{c} \right]^{-1}, \tag{37}$$

where $\mathbf{U}_n$ is the noise subspace, which is from the SPTFDs matrix of the selected T-F points. Selecting these points from high energy concentration regions can enhance the signal-to-noise ratio (SNR), which makes the proposed algorithm more robust than the conventional MUSIC counterparts. In (37), by obtaining the smallest eigenvalue of $\mathbf{F}^{\mathrm{H}}(\phi)\mathbf{U}_n\mathbf{U}_n^{\mathrm{H}}\mathbf{F}(\phi)$, the items in parentheses are the fewest. This algorithm can execute the simple $2 \times 2$ matrix eigendecomposition, thus avoiding abundant polarization dimensionality operations. Namely, the PTF-MUSIC spectrum can be simplified as

$$P(\phi) = \lambda_{\min}^{-1} \left[ \mathbf{F}^{\mathrm{H}}(\phi) \mathbf{U}_n \mathbf{U}_n^{\mathrm{H}} \mathbf{F}(\phi) \right], \tag{38}$$

where $\lambda_{\min}[\cdot]$ represents solving the smallest eigenvalue. The signal DOA estimation value is the angle coordinate corresponding to the spectrum's highest value. For each DOA $\phi_n$ of $N$ signals, $n = 1, 2, \ldots, N$, each signal polarization parameter can be estimated via

$$\widehat{\mathbf{c}}(\phi_n) = \mathbf{v}_{\min} \left[ \mathbf{F}^{\mathrm{H}}(\phi_n) \mathbf{U}_n \mathbf{U}_n^{\mathrm{H}} \mathbf{F}(\phi_n) \right], \tag{39}$$

where $\mathbf{v}_{\min}[\cdot]$ is the eigenvector for the minimal eigenvalue $\lambda_{\min}[\cdot]$.

For achieving rotational invariance in the array, the $M$-element array can be divided into two $M-1$-element overlapping subarrays. The first and second subarray include, respectively, the left-most and right-most $M-1$ antennas, as shown in Figure 2. In addition, the two subarrays' steering matrices of the identically polarized sensors are, respectively, $\mathbf{A}_1$ and $\mathbf{A}_2$. Namely,

$$\begin{bmatrix} \mathbf{A}_2 \mathbf{Q}^{[v]} \\ \mathbf{A}_2 \mathbf{Q}^{[h]} \end{bmatrix} = \begin{bmatrix} \mathbf{A}_1 \mathbf{Q}^{[v]} \\ \mathbf{A}_1 \mathbf{Q}^{[h]} \end{bmatrix} \mathbf{\Psi}, \tag{40}$$

where the rotation operation $\boldsymbol{\Psi}$ is described as

$$\boldsymbol{\Psi} = \mathrm{diag}\left[e^{-\mathrm{j}2\pi\frac{d}{\lambda}\sin(\phi_1)}, ..., e^{-\mathrm{j}2\pi\frac{d}{\lambda}\sin(\phi_N)}\right]. \tag{41}$$

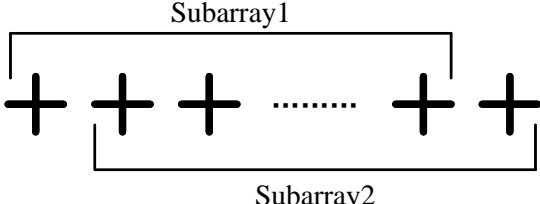

**Figure 2.** Two overlapping subarrays.

The signal eigenvectors consist of the columns of $\mathbf{U}_s$ acquired approximately from the signal subspace, which is spanned by the SPTFD matrices. A transformation matrix $\mathbf{T}$ satisfies

$$\mathbf{U_s} = \begin{bmatrix} \mathbf{A}\mathbf{Q}^{[v]} \\ \mathbf{A}\mathbf{Q}^{[h]} \end{bmatrix}\mathbf{T}. \tag{42}$$

By utilizing the same transformation matrix $\mathbf{T}$ to the two subarrays' steering matrices, $\mathbf{U}_{s1}$ and $\mathbf{U}_{s2}$ are described as

$$\mathbf{U_{s1}} = \begin{bmatrix} \mathbf{A}_1\mathbf{Q}^{[v]} \\ \mathbf{A}_1\mathbf{Q}^{[h]} \end{bmatrix}\mathbf{T}, \tag{43}$$

$$\mathbf{U_{s2}} = \begin{bmatrix} \mathbf{A}_2\mathbf{Q}^{[v]} \\ \mathbf{A}_2\mathbf{Q}^{[h]} \end{bmatrix}\mathbf{T}. \tag{44}$$

According to the above two equations,

$$\mathbf{U_{s2}} = \mathbf{U_{s1}}\mathbf{G}. \tag{45}$$

The matrix $\mathbf{G}$ satisfies

$$\mathbf{G} = \mathbf{T}\boldsymbol{\Psi}\mathbf{T}^{-1}, \tag{46}$$

where the matrix $\mathbf{G}$ eigenvalues are $e^{-\mathrm{j}2\pi\frac{d}{\lambda}\sin(\phi_n)}$, $n = 1, 2, \ldots, N$. For solving the matrix $\mathbf{G}$, the overdetermined Equation (46) is solved by the least squares approach and the total least squares approach. In this paper, the least squares approach is used:

$$\mathbf{G} = \left(\mathbf{U_{s1}^H}\mathbf{U_{s1}}\right)^{-1}\mathbf{U_{s1}^H}\mathbf{U_{s2}}. \tag{47}$$

At the low signal-to-noise ratio, the PTF-MUSIC algorithm has a higher success rate, but the estimation error is slightly larger. On the contrary, although the estimation error of the PTF-ESPRIT algorithm is small, the success rate of estimation is slightly lower. With the increase in SNR, the performance of both is improved. Similarly, PTF-MUSIC has the advantage of a high success rate, while PTF-ESPRIT has the advantage of small error. This paper observes these characteristics and combines them. Based on the rough estimation from PTF-MUSIC, the results obtained by PTF-ESPRIT are compared with the rough estimation results. If the difference between the two is within 5 degrees, the results of PTF-ESPRIT are selected. The success rate of estimation is guaranteed, and the accuracy of estimation is improved as much as possible.

## 5. Spatial Polarization Correlations

The array spatial resolution capability is primarily related to the signal propagation peculiarities. This lies in the normalized inner product of each array manifold vector. For spatial and polarimetric dimensions problems, the spatial polarization correlation

coefficient between the signal sources $l$ and $k$ is defined by the extended array manifold $\widetilde{\mathbf{a}}(\phi)$.

$$
\begin{aligned}
\beta_{l,k} &= \frac{1}{M} \widetilde{\mathbf{a}}^{\mathrm{H}}(\phi_k) \widetilde{\mathbf{a}}(\phi_l) \\
&= \frac{1}{M} \left( c_{k1}^* c_{l1} \left( \mathbf{a}^{[v]}(\phi_k) \right)^{\mathrm{H}} \mathbf{a}^{[v]}(\phi_l) + c_{k2}^* c_{l2} \left( \mathbf{a}^{[h]}(\phi_k) \right)^{\mathrm{H}} \mathbf{a}^{[h]}(\phi_l) \right) \\
&= c_{k1}^* c_{l1} \beta_{l,k}^{[v]} + c_{k2}^* c_{l2} \beta_{l,k}^{[h]} ,
\end{aligned}
\tag{48}
$$

where $\beta_{l,k}^{[i]} = (1/M) \left( \mathbf{a}^{[i]}(\phi_k) \right)^{H} \mathbf{a}^{[i]}(\phi_l)$ represents the spatial coefficient.

For the identical array manifolds between vertical and horizontal polarization, $\mathbf{a}^{[v]}(\phi) = \mathbf{a}^{[h]}(\phi)$ and $\beta_{l,k}^{[v]} = \beta_{l,k}^{[h]}$. The spatial polarization relation coefficient is the product of the individual spatial coefficient and polarimetric coefficient:

$$
\beta_{l,k} = \beta_{l,k}^{[v]} \rho_{l,k} ,
\tag{49}
$$

with

$$
\rho_{l,k} = \mathbf{c}_k^{\mathrm{H}} \mathbf{c}_l = \cos(\gamma_l)\cos(\gamma_k) + \sin(\gamma_l)\sin(\gamma_k) e^{j(\eta_l - \eta_k)},
\tag{50}
$$

expressing the polarimetric coefficient. Particularly, $\eta_{n_l} = \eta_{n_k} = 0$ (linear polarization) (50) simplifies to

$$
\rho_{l,k} = \cos(\gamma_l - \gamma_k).
\tag{51}
$$

Since $|\rho_{l,k}| \leq 1$, the equality is satisfied only when two signal polarization states are the same. The spatial polarization correlation coefficient is smaller than the single spatial coefficient. The introduced algorithm enhances signal recognition via dropping the correlation coefficient value with the polarization diversity. Namely, the signal recognition is difficult to obtain via a monopolar spatial array manifold, $\mathbf{a}^{[v]}(\phi)$ or $\mathbf{a}^{[h]}(\phi)$. However, the signals can be relatively easily identified by the extended spatial polarization array manifold, $\widetilde{\mathbf{a}}(\phi)$. This enhancement is more pronounced when there is a large spatial correlation coefficient and small spatial polarization correlation coefficient.

## 6. Simulations Results

According to the development of DOA estimation described in the introduction of this paper, some algorithms are summarized to compare the performance. The previous algorithm will not be elaborated in this paper. A total of nine algorithms are compared and analyzed in this section. Algorithm 1 is scalar MUSIC; Algorithm 2 is TF-MUSIC; Algorithm 3 is polarized MUSIC; Algorithm 4 is PTF-MUSIC; Algorithm 5 is the estimation of polarization DOA with eliminating strong interference. Algorithm 6 is TF-MUSIC based on cross terms. Algorithm 7 is a polarization ESPRIT based on time-frequency cross terms. Algorithm 8 is a polarization MUSIC based on time-frequency cross terms. Algorithm 9 is a dual-algorithm polarization DOA estimation based on time-frequency cross terms, which is the proposed algorithm in this paper. For convenience, these algorithms are referred to as A1~A9.

### 6.1. Time-Frequency Spectrum and Space Spectrum

Firstly, the cross terms extraction in time-frequency analysis was analyzed. The simulation parameters were as follows. The array number was 4, and the array spacing was half of a wavelength. The incidence angles of strong interference and weak signal were 30 and 60, respectively. The auxiliary polarization angles were 45° and 20°. The polarization phase differences were 0 degrees and 180 degrees. The extraction cross terms parameter was 0.01. The jamming-to-signal ratio (JSR) of strong interference was 7 dB. The decision threshold of the estimated result was 5 degrees. The normalized frequency of strong interference was 0.2 to 0.4. The weak signal normalized frequency was 0 to 0.2. The environment SNR was −4 dB. The time-frequency analysis algorithm was WVD. The snapshot number was 512.

The time-frequency spectrum of single-channel single-polarization is shown in Figure 3. The time-frequency spectrum after cross terms extraction is shown in Figure 4. It was impossible to obtain the weak nonstationary signal auto terms in the unprocessed time-frequency spectrum. There are two main reasons for this difficulty. First, from the point of view of position distribution, because the signal distribution is relatively concentrated, it is not easy to distinguish in the background of noise. Second, from the perspective of energy size, when the SNR is low, the signal energy is lower than the time-frequency spectrum noise bottom, and the weak signal auto terms are submerged in the noise, further aggravating the difficulty of resolving and extracting the weak signal auto terms. In Figure 4, the middle part of the red dotted line represents the cross terms. The T-F cross terms of the strong and weak nonstationary signals can be obtained by extracting cross terms processing. There were some noisy T-F points, but the cross terms' T-F points were clearly visible.

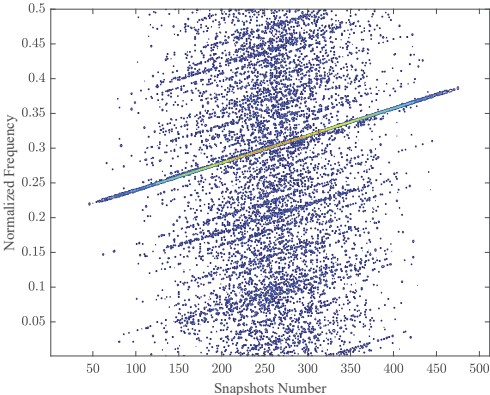

**Figure 3.** Unprocessed time-frequency spectrum.

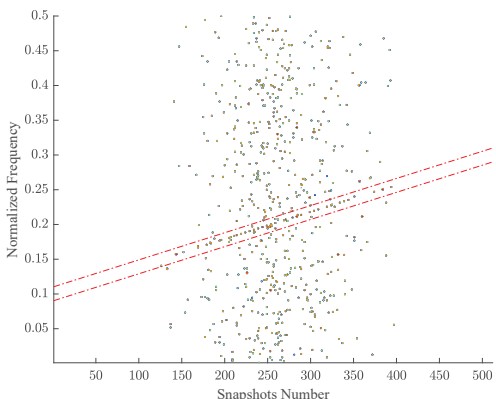

**Figure 4.** The time-frequency spectrum after extracting the cross terms.

The weak nonstationary signal DOA was estimated on the basis of cross terms, and the results are shown in Figure 5. It was almost difficult for A1~A6 to distinguish and estimate the weak signal, while A7 and A8 could obtain it. Among the weak signal results estimated by the algorithms in Figure 5, Algorithms 7 and 8 have the highest credibility, and they are also the basis for further estimation of DOA by the algorithm in this paper. The final estimation results of the algorithm in this paper refer to the results of Algorithms 7 and 8. Figure 5 shows the spatial spectrum of a single DOA estimate, which is only a qualitative analysis. The specific quantitative analysis of the algorithm proposed in this paper (Algorithm 9) is given by the following simulation analysis. The estimated success rate and specific accuracy were given by the following simulation.

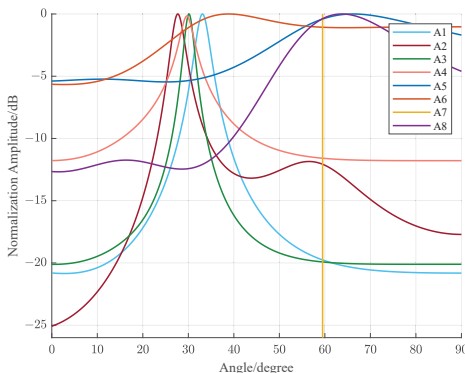

**Figure 5.** Comparison of weak signal results estimated by various algorithms.

## 6.2. Effect of SNR on Success Rate and RMSEs

The simulation conditions were the same as in Section 6.1. Among them, only the environment SNR changed from −10 dB to 10 dB, and the step size was 1 dB. The number of Monte Carlo cycles was 1000. The relationship curve between success rate and environment SNR is shown in Figure 6, and the relationship curve between the root-mean-square errors (RMSEs) and environment SNR is shown in Figure 7. With the increased environment SNR, the success rate increased and the RMSEs decreased. A8 and A9 always had better success rates than others, especially at low SNR. The RMSEs of A7 and A9 were consistently lower than those of others, especially at high SNR. A9 was overall optimal, with a high success rate and small RMSEs.

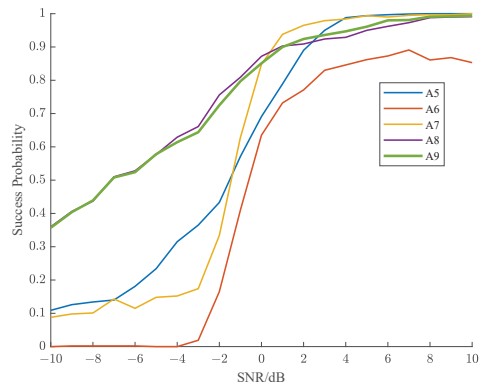

**Figure 6.** The relationship curve between success rate and SNR.

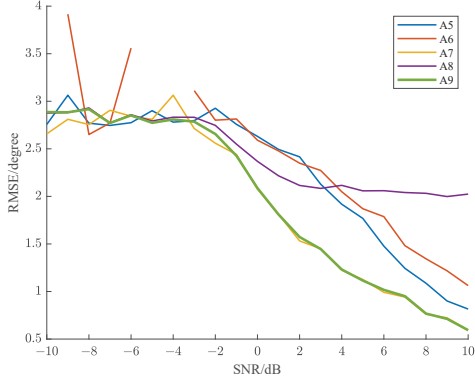

**Figure 7.** The relationship curve between RMSEs and environment SNR.

### 6.3. Effect of JSR on Success Rate and RMSEs

The simulation conditions were the same as in Section 6.1. Among them, only the JSR changed from 0 dB to 10 dB, and the step size was 1 dB. The relationship curve between success rate and JSR is shown in Figure 8, and the relationship curve between RMSEs and JSR is shown in Figure 9. With the increase in JSR, the success rate decreased, RMSEs increased, and A6 almost failed. Due to the lack of polarization information assistance, even in the cross terms, strong interference still affected the DOA estimation of weak signal. The success rates of A8 and A9 were always better than others, especially when the JSR was large. The RMSEs of A7 and A9 were always lower than others, especially when the JSR was small. A9 was overall optimal, with a high success rate and small RMSEs.

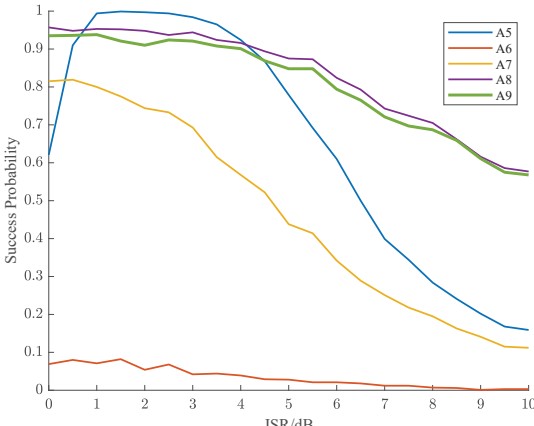

**Figure 8.** The relationship curve between success rate and JSR.

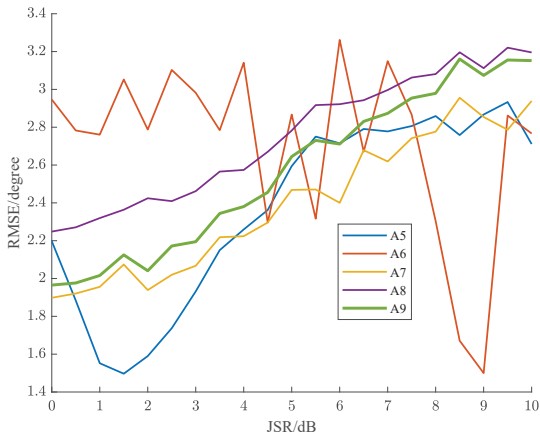

**Figure 9.** The relationship curve between RMSEs and JSR.

### 6.4. Effect of Snapshot Number on Success Rate and RMSEs

The simulation conditions were the same as in Section 6.1. Among them, the only snapshot numbers were 64, 128, 256, 512, and 1024. The relationship curve between success rate and snapshot number is shown in Figure 10, and the relationship curve between RMSEs and snapshot number is shown in Figure 11. With the increase in the snapshot number, the success rate increased and the RMSEs decreased. The success rates of A8 and A9 consistently outperformed the others. The RMSEs of A7 and A9 were always lower than the others, especially when the snapshot number was larger. A9 was overall optimal, with a high success rate and small RMSEs.

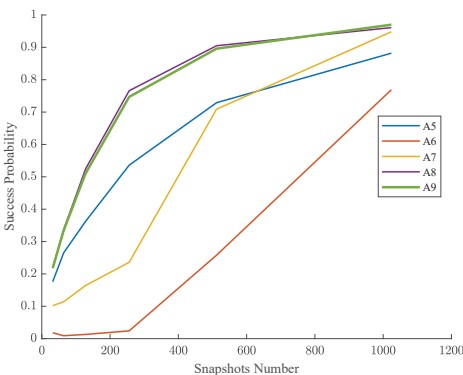

**Figure 10.** The relationship curve between success rate and snapshots number.

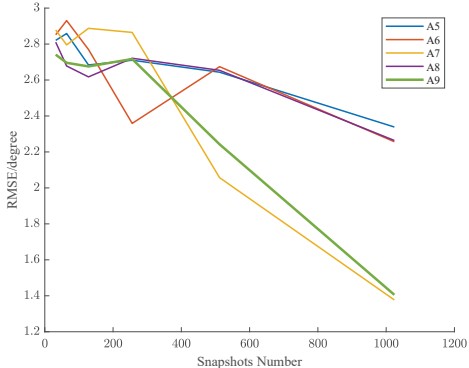

**Figure 11.** The relationship curve between RMSEs and snapshots number.

### 6.5. Effect of SNR and Epsilon on Success Rate and RMSEs

The simulation conditions were the same as in Section 6.1. Among them, the SNR changed from −10 dB to 10 dB, and the step size was 1 dB. The epsilons were 0.1, 0.05, and 0.01. The relationship curve between success rate and SNR and epsilon is shown in Figure 12, and the relationship curve between RMSEs and SNR and epsilon is shown in Figure 13. With the increase in SNR, the success rate increased and the RMSEs decreased. In the case of high SNR, the smaller the cross terms extraction parameter in A9, the higher the success rate and the smaller the RMSEs. At low SNR, the cross terms extraction parameter in A9 had little effect on success rate and RMSEs. In other algorithms, the cross terms extraction parameters had little effect on the success rate and RMSEs. A9 with small cross terms extraction parameters was the overall best, with a high success rate and small RMSEs.

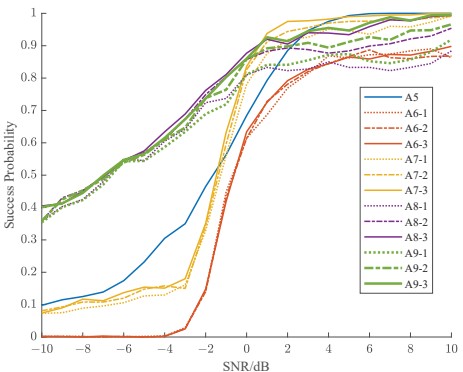

**Figure 12.** The relationship curve between success rate and SNR and epsilon.

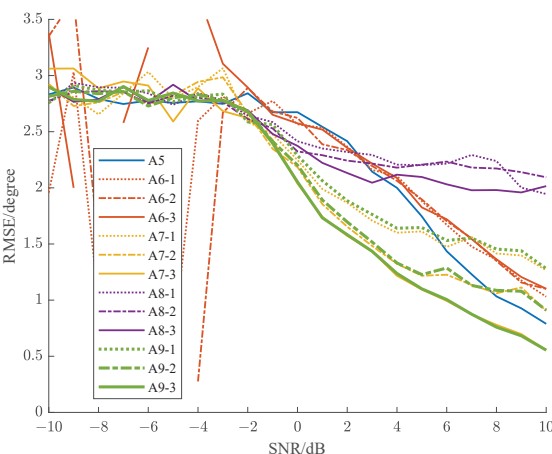

**Figure 13.** The relationship curve between RMSEs and SNR and epsilon.

### 6.6. Effect of Polarization Error on Success Rate and RMSEs

The simulation conditions were the same as in Section 6.1. Among them, the auxiliary polarization angle and polarization phase difference of polarization parameter error changed from 0 to 8 degrees, and the step size was 0.5 degrees, separately. The relationship curve between the success rate and the auxiliary polarization angle error is shown in Figure 14, and the relationship curve between RMSEs and the auxiliary polarization angle error is shown in Figure 15. The relationship curve between the success rate and the polarization phase difference error is shown in Figure 16, and the relationship curve between RMSEs and the polarization phase difference error is shown in Figure 17.

A6 and A7 had low success rates and almost failed, and RMSEs had no reference significance. A5 was almost unaffected by the estimation error of polarization parameters. Without the assistance of time-frequency information, A5 was still affected by residually strong interference, which weakened the parameters' estimation error influence. The success rate of A8 and A9 decreased, and the RMSEs increased with the increase in the auxiliary polarization angle estimation error. For the auxiliary polarization angle estimation error, A9 had the same approximate success rate as A8 and lower RMSEs than A8. For the polarization phase difference estimation error, A9 and A8 were almost identical. The effect of the polarization phase difference estimation error was little. It can be seen from $\mathbf{c}_n = [c_{nv}, c_{nh}]^{\mathrm{T}} = \left[\cos(\gamma_n), \sin(\gamma_n)e^{j\eta_n}\right]^{\mathrm{T}}$ that the polarization phase difference estimation error affected the exponential part and had little effect on solving the noise subspace of the SPTFDs matrix.

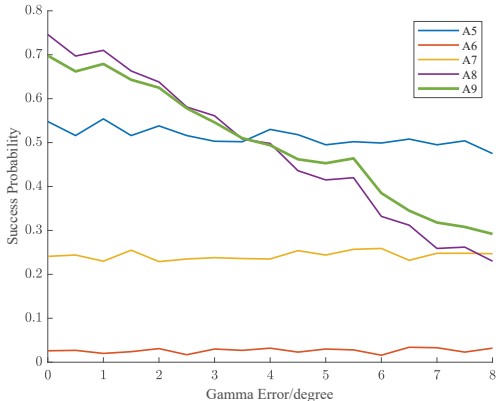

**Figure 14.** The curve between the success rate and the auxiliary polarization angle error.

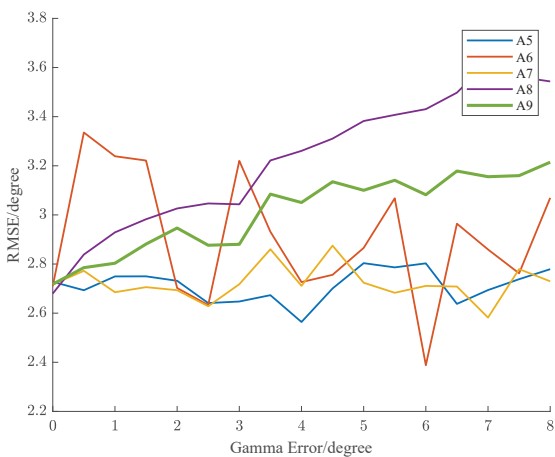

**Figure 15.** The curve between RMSEs and the auxiliary polarization angle error.

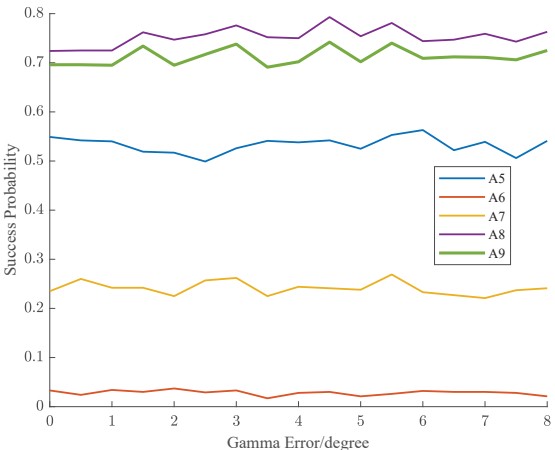

**Figure 16.** The curve between the success rate and the polarization phase difference error.

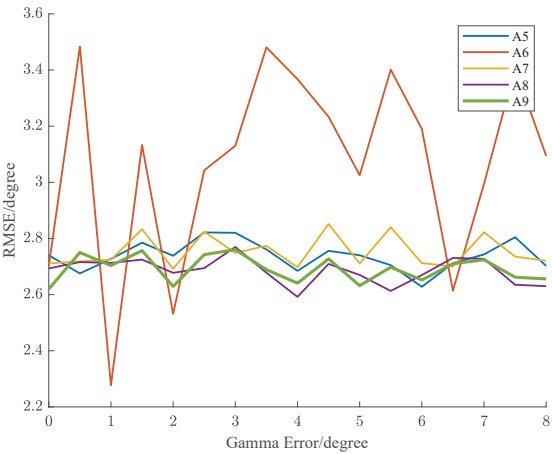

**Figure 17.** The curve between RMSEs and the polarization phase difference error.

All of the above simulations are summarized below. (1) The cross terms can be extracted. Algorithm 9 is feasible. (2) In many aspects, such as environmental SNR, JSR and snapshot number, A9 is the best overall, with a high success rate and small RMSEs. (3) The cross terms extraction parameters have little impact on the algorithm. When it is small, A9 has better performance. (4) Among the polarization parameter estimation errors,

the polarization auxiliary angle estimation error has a greater impact on the algorithm than the polarization phase difference estimation error. Even if there is an estimation error, A9 still performs well.

## 7. Conclusions

A platform for dealing with the DOA estimation of weak nonstationary polarization signals via utilizing T-F cross terms when both strong and weak signals coexist was proposed. This platform, termed SPTFDs based on cross terms, uses the spatial, polarization and T-F characteristics of the signals incident on the dual-polarized antenna array. The signal source separation is based on their DOA, their polarization and T-F peculiarities. The application of TFDs indicates the signal source time-varying frequency characteristics. It considers cross terms' T-F points with high energy concentrations when the weak nonstationary signal is submerged in noise. The eigendecomposition of the SPTFDs matrix from the T-F characteristics of the incident signals is utilized for defining the algorithm, which is polarization DOA estimation using dual algorithms based on T-F cross terms. The simulation results show that this algorithm is superior to the other existing DOA estimation algorithms for weak nonstationary signals when strong and weak nonstationary signals exist simultaneously.

**Author Contributions:** Conceptualization, S.S. and A.L.; methodology, S.S.; software, S.S.; validation, S.S., A.L. and C.Y.; formal analysis, S.S.; investigation, S.S., A.L. and X.W.; resources, X.W. and C.Y.; data curation, S.S. and X.W.; writing—original draft preparation, S.S.; writing—review and editing, S.S. and X.W.; visualization, S.S.; supervision, X.W. and C.Y.; project administration, A.L. and C.Y.; funding acquisition, A.L. All authors have read and agreed to the published version of the manuscript.

**Funding:** This research was funded by the Natural Science Foundation of Shandong Province (No. ZR2020MF013) and the National Natural Science Foundation of China (No. 62031015, No. 61971159).

**Institutional Review Board Statement:** Not applicable.

**Informed Consent Statement:** Not applicable.

**Data Availability Statement:** Data are available upon request from the corresponding author.

**Conflicts of Interest:** The authors declare no conflict of interest.

## Abbreviations

The following abbreviations are used in this manuscript:

| | |
|---|---|
| DOA | direction of arrival |
| T-F | time-frequency |
| STFDs | spatial time-frequency distributions |
| SPTFDs | spatial polarimetric time-frequency distributions |
| MUSIC | multiple signal classification |
| ESPRIT | estimation of signal parameters using rotational invariance technique |
| TFDs | time-frequency distributions |
| SAR | synthetic aperture radar |
| PTF-MUSIC | polarimetric time-frequency MUSIC |
| TEM | transverse electromagnetic |
| WVD | Wigner–Ville distribution |
| JAD | joint antidiagonalization |
| TF-MUSIC | time-frequency MUSIC |
| SNR | signal-to-noise ratio |
| JSR | jamming-to-signal ratio |
| RMSEs | root-mean-square errors |

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
