# Peer review of "Polarization Direction of Arrival Estimation Using Dual Algorithms Based on Time-Frequency Cross Terms"

_electronics, doi:10.3390/electronics12173575_

Round 1

Reviewer 1 Report

The paper presents a way of estimating week nonstationary polarization signal DOA by using spatial polarimetric TF distributions of cross terms when there are strong nonstationary polarization interference signals and additivity Gaussian white noise. The basic idea is interesting. The paper shows a detail theory and implementation. However, the main contribution can be more addressed detailly in the introduction. Besides, it would be good to compare the corresponding performance to other prior works.

Minor editing of English language required

Author Response

Dear Reviewers:

Thank you for your letter and for the comments concerning our manuscript entitled “Polarization Direction of Arrival Estimation Using Dual-Algorithms Based on Time-Frequency Cross Terms” (ID: 2526698). Those comments are all valuable and very helpful for revising and improving our paper, as well as the important guiding significance to our researches. We have studied comments carefully and have made correction which we hope meet with approval. Revised portion are marked in red in the paper. The main corrections in the paper and the responds to the reviewer’s comments are as following:

Point 1: The main contribution can be more addressed detailly in the introduction.

Response 1: Thank you for your comments. The main contributions are explained in more detail in the introduction and are marked in red font.

Point 2: It would be good to compare the corresponding performance to other prior works.

Response 2: Thank you for your comments. In this paper, several previous related algorithms are listed, among which the representative algorithms are algorithm 7 (polarization ESPRIT based on time-frequency cross terms) and algorithm 8 (polarization MUSIC based on time-frequency cross terms). This paper combines the characteristics of the two algorithms to increase the success rate and improve the accuracy. The performance of several algorithms and the factors affecting the performance are analyzed. Finally, it is concluded that the proposed algorithm has better performance for the scene with both strong and weak signals. Please see the simulation analysis section for details.

Point 3: Minor editing of English language required.

Response 3: Thank you for your comments. We have examine the whole article carefully again and further improved the writing of English.

Reviewer 2 Report

I am responding to MDPI request for me to provide you with a report in connection with the submitted manuscript “Polarization Direction of Arrival Estimation Using Dual-Algorithms Based on Time-Frequency Cross Terms” by Shuai Shao, Aijun Liu, Xiuhong Wang, and Changjun Yu. I am pleased to do so.

In this article, the authors have introduced a novelty means for estimating weak non-stationary polarization signal DOA by utilizing spatial polarimetric time-frequency distributions of cross terms when there are strong nonstationary polarization interference signals and additivity Gaussian white noise. They have concluded that the simulation results show that the used algorithm is superior to the other existing DOA estimation algorithms for the weak nonstationary signals when strong and weak nonstationary signals exist simultaneously. Their research topic is very interesting, the presentation is very good and their results appear to be correct. The only comments are the following:

1)     Is it possible the authors to make some comments for the used Manifolds? It could be helpful for the audience.

2)     What is the computational time for their simulations? Could they make any mention to faster algorithms e.g. genetic algorithms (SoftwareX 10, 100355, 2019)? As far as I know such algorithms have been employed in signal processing.

3)     Could they include any reference related to “cross terms extraction”?

4)     The authors have written “It was impossible to obtain the weak nonstationary signal auto terms in the unprocessed time-frequency spectrum.” They have not explained the reason of this difficulty. Am I right?

5)     In Figure 5: Comparison of weak signal results estimated by various algorithms. Which of these algorithms is most trustable?

6)     The newest reference is dated 2018. They have to renew the reference section.

.

Author Response

Dear Reviewers:

Thank you for your letter and for the comments concerning our manuscript entitled “Polarization Direction of Arrival Estimation Using Dual-Algorithms Based on Time-Frequency Cross Terms” (ID: 2526698). Those comments are all valuable and very helpful for revising and improving our paper, as well as the important guiding significance to our researches. We have studied comments carefully and have made correction which we hope meet with approval. Revised portion are marked in red in the paper. The main corrections in the paper and the responds to the reviewer’s comments are as following:

Point 1: Is it possible the authors to make some comments for the used Manifolds? It could be helpful for the audience.

Response 1: Thank you for your comments. After the formula (9) in the paper, the used manifolds are further elaborated, and the corresponding red font is specifically seen in the paper.

Point 2: What is the computational time for their simulations? Could they make any mention to faster algorithms e.g. genetic algorithms (SoftwareX 10, 100355, 2019)? As far as I know such algorithms have been employed in signal processing.

Response 2: Thank you for your comments. Because there are too many factors affecting the simulation calculation time, including the configuration of the device, the step size of the algorithm, etc., in different configuration cases, different system parameters will produce different calculation time, and the length of time is not specified in the paper.  About faster algorithms is the follow-up content of this research, which mainly includes two aspects of improvement: hardware and algorithm.  Hardware considerations will be followed by the application of GUA.  At present, the focus of the algorithm is to use the mathematical structure of correlation matrix to realize automatic recognition.  This algorithm involves many information dimensions, including space, time, frequency, and polarization.  The combination with faster algorithms requires more detailed research and larger exposition, which will be considered for subsequent research.

Point 3: Could they include any reference related to “cross terms extraction”?

Response 3: Thank you for your comments. In both the cross-term suppression and cross-term extraction studies, the related properties of cross-term are discussed, which are valuable for the study of cross-term extraction.  It is convenient to study the cross-term of time-frequency distribution of spatial polarization from multiple angles, and provides theoretical basis for subsequent DOA.

Point 4: The authors have written “It was impossible to obtain the weak nonstationary signal auto terms in the unprocessed time-frequency spectrum.” They have not explained the reason of this difficulty. Am I right?

Response 4: Thank you for your comments. We further explain the reasons for this difficulty, which can be seen in the red font above Figure 4 in the article.

Point 5: In Figure 5: Comparison of weak signal results estimated by various algorithms. Which of these algorithms is most trustable?

Response 5: Thank you for your comments. We further provide the reliability analysis of the estimated results, which can be seen in the red font at the top of Figure 5 of this paper.

Point 6: The newest reference is dated 2018. They have to renew the reference section.

Response 6: Thank you for your comments. We further renew the reference section, adding 6 new references from the last three years. For details, please see the red part of the reference.

Point 7: Minor editing of English language required.

Response 7: Thank you for your comments. We have examine the whole article carefully again and further improved the writing of English.
